# Immunogenicity of a Heterologous Prime-Boost COVID-19 Vaccination with mRNA and Inactivated Virus Vaccines Compared with Homologous Vaccination Strategy against SARS-CoV-2 Variants

**DOI:** 10.3390/vaccines10010072

**Published:** 2022-01-03

**Authors:** Ruiqi Zhang, Danlei Liu, Ka-Yi Leung, Yujing Fan, Lu Lu, Pui-Chun Chan, Kelvin Kai-Wang To, Honglin Chen, Kwok-Yung Yuen, Kwok-Hung Chan, Ivan Fan-Ngai Hung

**Affiliations:** 1Department of Medicine, Li Ka Shing Faculty of Medicine, University of Hong Kong, Hong Kong Special Administrative Region, Hong Kong 999077, China; zhangrq@hku.hk (R.Z.); danlei6@connect.hku.hk (D.L.); jyjfan@connect.hku.hk (Y.F.); 2Department of Microbiology, Li Ka Shing Faculty of Medicine, University of Hong Kong, Hong Kong Special Administrative Region, Hong Kong 999077, China; joy2u@connect.hku.hk (K.-Y.L.); u3003963@connect.hku.hk (L.L.); bcpc@hku.hk (P.-C.C.); kelvinto@hku.hk (K.K.-W.T.); hlchen@hku.hk (H.C.); kyyuen@hku.hk (K.-Y.Y.); chankh2@hku.hk (K.-H.C.); 3State Key Laboratory for Emerging Infectious Diseases, Li Ka Shing Faculty of Medicine, University of Hong Kong, Hong Kong Special Administrative Region, Hong Kong 999077, China; 4Carol Yu Centre for Infection, Li Ka Shing Faculty of Medicine, University of Hong Kong, Hong Kong Special Administrative Region, Hong Kong 999077, China

**Keywords:** COVID-19, heterologous vaccination, variants, neutralizing antibody

## Abstract

The emergence of SARS-CoV-2 variants may impact the effectiveness of vaccines, while heterologous vaccine strategy is considered to provide better protection. The immunogenicity of an mRNA-inactivated virus vaccine against the SARS-CoV-2 wild-type (WT) and variants was evaluated in the study. SARS-CoV-2 naïve adults (*n* = 123) were recruited and placed in the following groups: BNT162b2, CoronaVac or BNT162b2-CoronaVac (Combo) Group. Blood samples were collected to measure neutralization antibodies (NAb) by a live virus microneutralization assay (vMN) and surrogate NAb test. The day 56 vMN geometric mean titre (GMT) was 26.2 [95% confident interval (CI), [22.3–30.9] for Combo, 136.9 (95% CI, 104.2–179.7) for BNT162b2, and 14.7 (95% CI, 11.6–18.6) for CoronaVac groups. At 6 months post-first dose, the GMT declined to 8.0, 28.8 and 7.1 in the Combo, BNT162b2 and CoronaVac groups, respectively. Three groups showed reduced neutralizing activity against D614G, beta, theta and delta variants. At day 56 GMT (74.6) and month 6 GMT (22.7), the delta variant in the BNT162b2 group was higher than that in the Combo (day 56, 7.4; month 6, 5.5) and CoronaVac groups (day 56, 8.0; month 6, 5) (*p* < 0.0001). Furthermore, the mean surrogate NAb value on day 56 in the BNT162b2 group was 594.7 AU/mL and higher than 40.5 AU/mL in Combo and 38.8 AU/mL in CoronaVac groups (*p* < 0.0001). None of the participants developed severe adverse events, and all other adverse events were self-limiting. The Combo vaccination strategy was safe. The overall vaccine immunogenicity at day 56 and 6 months were comparable to the homologous CoronaVac group but inferior to the homologous BNT162b2 group, against both the WT and all variants. Furthermore, the antibody response of vaccines waned at 6 months and thereby, a third dose of the vaccine is needed for these vaccines.

## 1. Introduction

The ongoing coronavirus disease 2019 (COVID-19) pandemic has lasted more than one and half years and continues to threaten the world. There are more than 250 million confirmed COVID-19 cases, including 5.2 million deaths [1]. COVID-19 is caused by the severe acute respiratory syndrome coronavirus 2 (SARS-CoV-2), and the vaccine is considered an effective way to contain the COVID-19 pandemic.

To date, there are seven vaccines recommended by World Health Organization (WHO) for emergency use, including four types of COVID-19 vaccines: mRNA, inactivated virus, viral vector-based and recombinant protein vaccine. BNT162b2, as the first mRNA vaccine approved by WHO, has shown high effectiveness and safety in clinical trials [1,2]. As a representative of inactivated virus vaccines, CoronaVac (Sinovac Life Sciences, Beijing, China) showed qualified immunogenicity in the phase 3 trial [3]. ChAdOx1(AZD1222) is a type of viral vector-based vaccine which looks to insert the SARS-CoV-2 spike protein gene into the chimpanzee adenovirus vector ChAdOx1 for replication, providing 70.4% vaccine effectiveness [4]. Although the recombinant protein vaccine NVX-CoV2373 (Novavax) has not been included in the emergency use list, it has been shown to be 89.7% effective in the clinical trial, and the efficacy against variant B.1.1.7 was approximately 86.3% [5].

However, with the spread of the pandemic, the accumulation of mutations induces the emergence of several variants, which impact the effectiveness of COVID-19 vaccines [6,7,8]. A variant that detected a single mutant with D614G in the spike protein has shown a higher ability of transmission and replication [9,10]. Subsequently, several major variants appeared in some severely affected areas and spread rapidly around the world, including the B.1.1.7 (Alpha) variant found in the United Kingdom (UK), B.1.351 (Beta) variant found in South Africa, P.1 (Gamma) variant found in Brazil and B.1.617.2 (Delta) in India. These variants accumulated multiple mutations and involved the spike protein [10,11]. Since most vaccine development uses the spike protein as an antigen, the mutations of the SARS-CoV-2 in the spike protein make the efficacy of the vaccine a concern. Some studies have examined the protection of vaccinations against variants in a population. The results showed that the vaccine is still effective against the current variants, but its effectiveness is weakened in the infected patients [10,12,13,14]. The mutation of the virus makes the prevention situation serious, and changes in vaccine strategies will continue to be monitored.

Meanwhile, the current situation of the shortage of vaccines in the world cannot be ignored. As an alternative, the possibility of sequentially administering different SARS-CoV-2 vaccines, known as heterologous schedules, could be an opportunity to make vaccination programs more flexible and reliable in the face of supply fluctuations. The strategy of heterologous prime-boost is considered to combine the best characteristic of each vaccine to enhance the immune response and may show better protection when the virus mutates; this kind of strategy has been used to protect against Ebola [15,16]. Just as some countries have recommended the use of mRNA vaccines for heterologous booster immunization after initial immunization with the ChAdOx1 vaccine, it has been proven to be a safe and effective strategy in the clinical trial [17].

The aim of this study was to explore the immunogenicity of a heterologous prime-boost COVID-19 vaccination with mRNA and inactivated virus vaccines and compare it with homologous vaccination. The result may provide new thinking for the current vaccine situation.

## 2. Materials and Methods

### 2.1. Study Design and Participants

This study was an open trial completed in two vaccination centres in Hong Kong. Healthy adults aged 18–60 years with no related infectious history were invited and recruited into the study. We informed the participants of the related precautions before the study and obtained their informed consent. The study was approved by the institutional review board of the University of Hong Kong and Hospital Authority (UW 21-214).

### 2.2. Procedures

The study was divided into three groups; participants in each group were vaccinated with different types of SARS-CoV-2 vaccine, including the BNT162b2, CoronaVac, and Combo (BNT162b2- CoronaVac) vaccine. In the BNT162b2 group, first and second doses of COVID-19 vaccines were vaccinated at day 0 and day 21 with the homologous vaccine, while the CoronaVac groups were vaccinated on day 0 and day 28. In the Combo vaccine group, participants received heterologous BNT162b2 and CoronaVac vaccines on day 0 and day 28, respectively. The study did not set a random mechanism and did not blind the participants and investigators. The participants could choose the type of vaccination according to their personal wishes and were notified of the collection of blood at baseline, day 21 (BNT162b2) or day 28 (CoronaVac and Combo vaccine), day 56 and month 6 (Figure 1).

A live virus microneutralization (vMN) assay was performed in the Biosafety Level 3 facility at HKU to compare the effectiveness of antibody levels against different SARS-CoV-2 strains, including the HKU-001a (wild type, GenBank accession number MT230904) strain (WT), B.1.36.27 (D614G), B.1.1.7 (alpha variant), B.1.351 (beta variant), P.3. (theta variant) and B.1.617.2 (delta variant) [18]. The serum samples were treated at 56 °C for 30 min before testing. VeroE6 TMPRSS2 cells (JCRB Cell Bank Catalogue no. JCRB1819) were seeded in a 96-well plate and incubated at 37 °C and 5% CO_2_. A serial two-fold dilution was performed and started at 1:10 for the serum. One hundred 50% tissue culture infective dose (TCID50) of SARS-CoV-2 virus was mixed with serum and incubated for 1.5 h at 37 °C, the mixture was added to the cells and placed in the 37 °C for 72 h incubation. Cytopathic effect (CPE) was observed and recorded at 72 h. The titre of vMN was defined as the maximum dilution of serum in which the percentage of CPE is equal to or less than 50%. vMN titre ≥10 was considered as positive.

Surrogate neutralizing antibody (NAb) against the SARS-CoV-2 RBD and anti-nucleocapsid protein (N) IgG was determined with chemiluminescent microparticle immunoassay on the iFlash 1800 analyzer (Shenzhen YHLO Biotech Co., Ltd., Shenzhen, China) [18]. In the sample loading area, sera were placed on a sample rack, and then the iFlash system performed the test automatically. For surrogate NAb, the cut-off value for seropositivity was 15 AU/mL, and the maximum measurable value was 800 AU/mL. For the anti-N IgG, the cut-off value for seropositivity was 10 AU/mL

### 2.3. Outcomes

The primary end-point was titre. The secondary end-point was geometric mean titre (GMT) fold increase and the rate of adverse events. For assessing safety, participants were informed of the vaccine name, risks and precautions and were asked to observe severe adverse events for half an hour after vaccination. All participants were given a symptom diary to record the adverse effects 4 weeks after each dose. The adverse events contain system reactions and local reactions. Systemic reactions included fever, headache, muscle pain, skin rash, joint pain, vomit, tiredness, diarrhea, chills and severe adverse events (SAE). SAE were defined as vaccine-related undesired events, including death, disability or life-threatening conditions. Pain at the injection site, itching, swelling, redness, and ecchymosis were listed as local reactions.

### 2.4. Statistics

Statistical inference of continuous variables was performed using one-way ANOVA and analysis of variance, least-significant difference (LSD) multiple comparison was used when meeting the homogeneity of variance, otherwise Dunnett’s multiple comparison was used, including demographic parameters and antibody level. Pearson chi-square and Fisher’s exact test were utilized for numerical or categorical data. SPSS 27.0 was used for statistical computation. Results all used a two-tailed test and were considered significant at *p* < 0.05.

## 3. Results

### 3.1. Subjects

Between May and November 2021, 122 SARS-CoV-2 naïve individuals were invited to the participate in the COVID-19 vaccination programme. Forty-two individuals received one dose of BNT162b2 followed by one dose of CoronaVac (median age = 44.5 years), 40 individuals received two doses of BNT162b2 (median age = 49 years) and 41 individuals received two doses of CoronaVac (median age = 47 years) (Table 1).

### 3.2. Immunogenicity of Vaccines Determined by vMN

The results of NAb in sera determined by vMN showed that these participants had no antibodies against WT on the baseline. On day 28, individuals from the Combo group had a higher geometric mean titre (GMT) [23.7, 95% confidence interval (CI), 18.8–29.9] against WT than CoronaVac groups (5.6, 95% CI, 5.1–6.2) (*p* < 0.0001) (Table 2) (Figure 2a). In the BNT162b2 group, day 21 vMN GMT was 13.4 (95% CI, 9.7–18.7) (Table 2). The vMN GMT on day 56 increased to 26.2 (95% CI, 22.3–30.9), 136.9 (95% CI, 104.2–179.7) and 14.7 (95% CI, 11.6–18.6) in the Combo, BNT162b2 and CoronaVac groups, respectively (Table 2). The GMT fold increase on day 56 was 5.2 (95% CI, 4.5–6.2) in Combo, 27.4 (95% CI, 20.8–36.0) in BNT162b2, and 2.9 (95% CI, 2.3–3.7) in the CoronaVac group. On day 56, GMT in the BNT162b2 group was significantly higher than that in the Combo (*p* < 0.0001) and CoronaVac groups (*p* < 0.0001), while no statistically significant difference was observed in GMT between the Combo and CoronaVac groups (Figure 2a). At 6 months post-first dose of vaccine, the GMT declined compared with that on day 56 and were 8.0 (95% CI,6.6–9.8), 28.8 (95% CI, 22.3–37.2) and 7.1 (95% CI, 4.8–10.5) in the Combo, BNT162b2 and CoronaVac groups, respectively (Table 2).

Besides WT, vMN titres against variants (D614G, alpha, beta, theta, delta) were also determined. For the D614G variant, the BNT162b2 group showed significantly higher results on day 56 GMT (81.4, 95% CI, 60.0–110.5) (*p* < 0.0001) than the Combo (13.1, 95% CI, 10.7–16.1) and CoronaVac groups (11.8, 95% CI, 9.6–14.5) (Table 2) (Figure 2b). At 6 months, the GMT against D614G in the Combo, BNT162b2 and CoronaVac groups reduced to 5.7 (95% CI, 5.0–6.4), 21.5 (95% CI, 17.0–27.2) and 5 (95% CI, 5–5), respectively. After full vaccination, there was no statistically significant difference in GMT at day 56 and month 6 between the Combo and CoronaVac groups (Figure 2b). For the alpha variant, individuals in the BNT162b2 group showed higher GMT on day 56 (146.7) (*p* < 0.0001) and month 6 (37.9) (*p* < 0.0001) than these from the Combo group (day 56 GMT, 11.5; month 6 GMT, 6.6) and the CoronaVac group (day 56 GMT, 11.6; month 6 GMT, 7.1), while the Combo group showed a comparable antibody response at day 56 and month 6 with CoronaVac group (Table 2) (Figure 2c).

Vaccines showed low effectiveness against the beta variant. On day 56, the GMT was 5.4 (95% CI, 4.8–5.9) against the beta variant in the Combo group, and 15.7 (95% CI, 12.0–20.5) in BNT162b2 and 5 (95% CI, 5–5) in the CoronaVac group (Table 2). The GMT on day 56 in the BNT162b2 group was higher than that in the Combo (*p* < 0.0001) and CoronaVac (*p* < 0.0001) groups and no statistically significant difference was observed in GMT between the Combo and CoronaVac groups (Figure 2d). These vaccination strategies also showed low effectiveness against the theta variant, and the day 56 GMT in the Combo group was 5.9 (95% CI, 5.2–6.8), which was lower than that in BNT162b2 (day 56 GMT, 39.3; 95% CI, 27.6–55.9) (*p* < 0.0001) and comparable with the CoronaVac group (day 56 GMT, 7.1; 95%CI, 5.9–8.6) (Table 2) (Figure 2e). For the delta variant, the day 56 GMT and month 6 GMT in the BNT162b2 group (day 56, 74.6; month 6, 13) was higher than that in the Combo (day 56, 7.4; month 6, 5.5) and CoronaVac groups (day 56, 8.0; month 6, 5) (*p* < 0.0001) (Table 2) (Figure 2f). When comparing the immunogenicity against delta between the Combo and CoronaVac groups, no statistically significant difference was observed at day 56 and month 6 (Figure 2f). Overall, the BNT162b2 platform showed the highest immunogenicity, and the Combo platform induced comparable antibody response with CoronaVac groups against WT and five variants.

### 3.3. Immunogenicity of Vaccines Determined by Surrogate NAb Test

In addition to the vMN assay, chemiluminescent microparticle immunoassay, which is a type of surrogate NAb test, was used to determine the antibody response induced by vaccines. Similar to the results produced by vMN, the mean surrogate NAb value was 37.1 AU/mL for the Combo group, 70.6 AU/mL for the BNT162b2 group, and 5.5 AU/mL for the CoronaVac group post first dose of vaccine (Table 3). On day 56, individuals from the Combo group showed a slight increase in the surrogate Nab value (mean, 40.5 AU/mL), while the mean of the surrogate NAb values in the other two groups jumped to 594.7 AU/mL (BNT162b2) and 38.8 AU/mL (CoronaVac). Then, at 6 months, there was a decrease of the surrogate NAb level, and the mean was 8.6 AU/mL in Combo, 138.5 AU/mL in BNT162b2 and 9.2 AU/mL in the CoronaVac group (Table 3). Furthermore, anti-N IgG was also tested. Only the CoronaVac group showed anti-N IgG on day 56, and the mean value was 21.8 AU/mL (Table 3).

### 3.4. Safety

After a primer dose of vaccine, the incidence of total local symptoms in the BNT162b2 and Combo groups was significantly higher than that of CoronaVac (Combo, 97.6%; BNT162b2, 81.8%; CoronaVac, 34.3%; *p*< 0.0001), and there was no significant difference in total systemic reactions between three groups (Combo, 45.2%; BNT162b2, 45.5%; CoronaVac, 37.1%; *p* = 0.761). The second dose of the vaccine induced higher frequencies of systemic reactions (66.7%) (*p* < 0.0001) and local reactions (75.8%) (*p* < 0.0001) in the BNT162b2 group than in the Combo (systemic reactions, 31.0%; local reactions, 35.7%) and CoronaVac groups (systemic reactions, 22.9%; local reactions, 20.0%) (Table 4). The most common symptom induced by the second dose of COVID-19 vaccine in three groups was injection site pain (Combo, 35.7%; BNT162b2, 69.7%; CoronaVac, 20.0%). After the second dose, the frequencies of chill, tiredness, muscle pain, joint pain, redness, swelling and itching in the BNT162b2 group were higher than those in the Combo and CoronaVac groups (Table 4). No SAE was observed in the study after vaccination.

## 4. Discussion

In this study, the efficacy of three vaccination strategies against SARS-CoV-2 was determined. In the vMN assay with WT, the Combo group showed higher immunogenicity than other groups after the first dose, while the BNT162b2 platform showed the highest immunogenicity after full vaccination. When the serum samples were tested again in different variants, we found that the antibody titers reduced, especially against the beta and theta variants. After 6 months of vaccination, the neutralizing activity dropped further in the WT and all variants.

Comparing the immunogenicity of the three strategies against WT, BNT162b2 showed significantly higher efficacy after full vaccination (day 56 GMT, 136.9), and the serum samples of subjects vaccinated with the Combo vaccine (day 56 GMT, 26.2) also showed higher efficacy than CoronaVac (day 56 GMT, 14.7) (Table 1). The development of BNT162b2 and CoronaVac is based on two different platforms. The CoronaVac vaccine involves inactivated viruses to activate humoral immunity, while the BNT162b2 vaccine works via translating targeted antigens to induce an immune response in the host [19,20,21]. As the immunogenicity of vaccines from various platforms is different after vaccination, heterogeneous prime-boost vaccination that combines different platforms may produce comparable efficacy with, or even higher efficacy than, the homologous CoronaVac vaccine [2,3]. In one clinical study, Vasileiou et al. reported that the effectiveness of primer-dose BNT162b2 was 85% at day 28–34 post-vaccination and then dropped to a level of 64% after 42 days post-vaccination [22]. Furthermore, from our previous study, the neutralizing activity in recovered patients with previous COVID-19 infection decreased significantly at week 6 after discharge [23]. Thus, the antibody response on day 56 might be lower than on day 28 in healthy individuals who have just received one dose of BNT162b2. In this study, the day 56 GMT was comparable to the day 28 GMT in the Combo group. Despite an inferior neutralizing antibody response when compared to the homologous BNT162b2, one dose of the CoronaVac vaccine could maintain the immune response induced by the primer dose of BNT162b2 in the Combo group. Studies on the mRNA vaccines have demonstrated their high clinical effectiveness and are adopted widely throughout the world [24,25]. However, the shortage of mRNA COVID-19 vaccines in some countries resulted in the delay of the second dose [26]. For these individuals who cannot receive the second dose of the BNT162b2 vaccine on time, they can opt for one dose of CoronaVac to complete the vaccination regime. Furthermore, subjects who cannot tolerate the side effects induced by the second dose of the BNT162b2 vaccine can also switch to one dose of CoronaVac after priming with the BNT162b2 (Table 4).

The various mutation of the SARS-CoV-2 has caused concerns regarding the effectiveness of the current vaccines; therefore, we have tested antibodies against the main variants. The results showed that the vMN titre all decreased for the D614G, beta, theta and delta variants. Compared with Combo and CoronaVac vaccine, BNT162b2 still showed better immunogenicity against the variant virus after two doses. The Combo vaccine has not elicited better neutralizing activity against variants than the CoronaVac. Interestingly, a reduction of GMT on day 56 and month 6 against alpha was observed in the Combo and CoronaVac groups when compared with that against WT, but the neutralizing activity against alpha and WT was similar after two doses of BNT162b2 (Table 2). The reason may be that mutation in the alpha variant caused the escape of the virus from the antibody response induced by the CoronaVac vaccine [27,28]. It has been reported that 31.8% and 59.1% of plasma from individuals receiving the CoronaVac vaccine failed to detect the effective neutralization of alpha and gamma variants. Furthermore, our study found that the performance of these vaccines in beta and theta variants was inferior to other variants. Both the beta and theta variants contain N501Y and E484K mutations in the S-protein, which enhanced the affinity of ACE2, and immune sera showed decreased activity when the mutations, including N501Y and E484K [6,10,29]. Although the delta variant is more contagious, BNT162b2 still showed satisfactory efficacy after two doses (day 56 GMT, 74.6). A study has shown that the effectiveness of two doses of BNT162b2 was 88.0% among the vaccinated population with the delta variant, while 93.7% among those with alpha variant, but the effectiveness against the delta variant was significantly lower than that of the alpha variant after one dose [30].

Interestingly, the vMN test for WT performed on the participants before the second dose showed that the vMN titer of the Combo group was higher than that of the BNT162b2 group (23.7 vs. 13.4), even though the primer dose of the two groups were the same. The difference lies in the interval between vaccination. The Combo group assay was taken on day 28, while the BNT162b2 group was on day 21. It may indicate that after the first injection of the mRNA vaccine, the immunogenicity has not reached the peak value in 21 days; thereby the delay of the second dose of the BNT162b2 vaccine may have a better antibody response. Currently, there is no comparative study on the vaccine interval of BNT162b2, but in the ChAdOx1 nCoV-19 vaccine study, it has shown higher efficacy with a vaccination interval over six weeks [4]. Meanwhile, delaying the second dose might offer a good strategy to tackle vaccine shortages and could expand the vaccine coverage, especially in the high-risk population [31].

Moreover, the neutralizing protection induced by vaccines in three groups dropped a lot at 6 months when compared to that on day 56. For delta, month 6 GMT decreased from 7.4 to 5.5 in the Combo group, from 74.6 to 22.7 in the BNT162b2 group, and from 8.0 to 5 in the CoronaVac group (Table 2). For WT, the month 6 GMT was only 8.0, 28.8 and 7.1 in the Combo, BNT162b2, and CoronaVac groups, respectively. Some studies also reported a reduction in the effectiveness of COVID-19 at 6 months [7,8]. One clinical study conducted in the USA reported that the BNT162b2 vaccine effectiveness against delta decreased to 53% after 4 months when compared to 93% during the first month after two doses of vaccine [7]. Thus, the booster dose should be administered to these individuals receiving BNT162b2 at 5 months post-second dose, while people vaccinated with the BNT162b2-CoronaVac combination or CoronaVac should receive the third dose earlier. The BNT162b2 vaccine would be the preferred COVID-19 booster, according to these study results.

The limitation of the study is the lack of sera collected between day 56 and month 6, and as a result, could not determine the optimal time to receive the third dose of vaccine. The time of the sera collection after the first dose was different between the BNT162b2 group and the Combo and CoronaVac groups due to the time restriction for the second dose for the respective vaccine platforms. We were unable to test the various vaccine platforms, especially the Combo platform, in preventing COVID-19 infection, as there were very few imported COVID-19 patients and no local patients over the previous 6 months due to the very strict quarantine policy imposed on the returned travelers. Besides, the immunogenicity of CoronaVac as a primer followed by one dose of the BNT162b2 vaccine deserves further investigation, and we have began to recruit volunteers to receive the heterologous CoronaVac-BNT162b2 vaccination strategy.

## 5. Conclusions

In this study, heterologous vaccination strategy was safe. The overall vaccine immunogenicity of the Combo platform at day 56 and 6 months was inferior to the homologous BNT162b2 platform but comparable to the homologous CoronaVac platform, against both WT and all variants. Furthermore, the neutralizing activity against SARS-CoV-2 waned at 6 months regardless of the three vaccination platforms. Thus, it is recommended that one booster of COVID-19 vaccine should be administered at 5 months after completing the second dose. Furthermore, the mix-and-match approach for the booster shot requires further investigation regarding people who have already received two doses of the COVID-19 vaccines.

## Figures and Tables

**Figure 1 vaccines-10-00072-f001:**
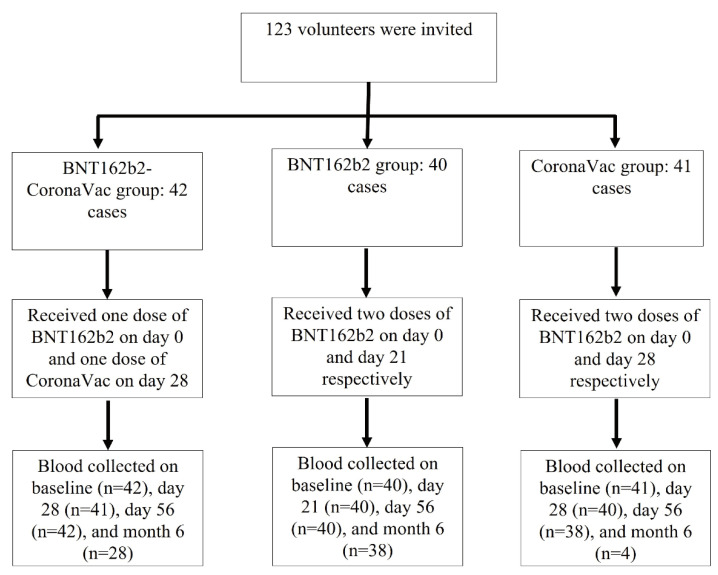
Participant recruitment and research flow diagram.

**Figure 2 vaccines-10-00072-f002:**
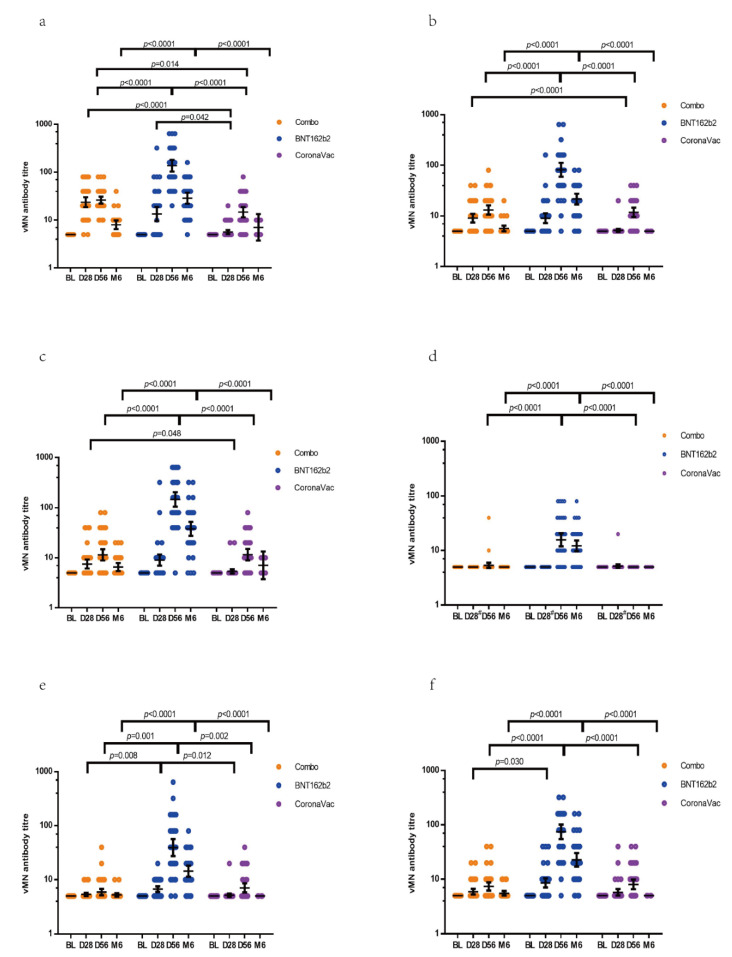
Comparison of immunogenicity of three vaccination strategies. SARSA-CoV-2 naïve individuals were invited to participate in the vaccination programme. In the Combo group, subjects received one dose of BNT162b2 on day 0 and one dose of CoronaVac on day 28. Then, blood samples were collected on the baseline, day 28, day 56 and month 6. For BNT162b2, subjects recieved two doses of BNT162b2 at the baseline and day 21 respectively and blood was taken at the baseline, day 21, day 56 and month 6. For CoronaVac, Individuals received two doses of CoronaVac vaccine at the baseline and day 28, respectively, and then blood was taken at the baseline, day 28, day 56 and month 6. The live virus microneutralization assay (vMN) was used to determine the level of neutralizing antibodies in serum against (**a**) wild type, (**b**) D614G, (**c**) alpha, (**d**) beta, (**e**) theta and (**f**) delta. BL: baseline; D28: day 28 for Combo and CoronaVac, day 21 for BNT162b2; D56: day 56; M6: month 6. To analyse the difference in immunogenicity between two groups, a post hoc multiple comparison was performed, and a *p* value was shown on the graph if the value was less than 0.05, which represents a statistically significant difference.

**Table 1 vaccines-10-00072-t001:** Demographic and clinical baseline characteristics.

	Combo (*n* = 42)	CoronaVac (*n* = 41)	BNT162b2 (*n* = 40)	*p* ^2^
Median age (years)	44.5 (36–50.5) ^1^	49 (39.5–54.5)	47 (33–51.75)	0.232
Male/Female (%)	24/18 (57.1/42.8)	13/28 (31.7/68.3)	19/21(47.5/52.5)	0.065

^1^ data were median age (IQR). ^2^ *p* < 0.05, the results are significantly different.

**Table 2 vaccines-10-00072-t002:** Immunogenicity of different vaccination strategies.

	Combo (*n* = 42)	BNT162b2 (*n* = 40)	CoronaVac (*n* = 41)	*p* ^9^
**WT**				
**Baseline**				
GMT ^1^	5 (5–5)	5 (5–5)	5 (5–5)	
**Day 21 ^2^/Day 28**				
GMT	23.7 (18.8–29.9)	13.4 (9.7–18.7)	5.6 (5.1–6.2)	0.002
GMT fold increase	4.7 (3.8–6.0) ^3^	2.7 (1.9–3.7)	1.1 (1.0–1.2) ^6^	0.002
**Day 56**				
GMT	26.2 (22.3–30.9)	136.9 (104.2–179.7)	14.7 (11.6–18.6)	<0.0001
GMT fold increase	5.2 (4.5–6.2)	27.4 (20.8–36.0)	2.9 (2.3–3.7) ^7^	<0.0001
**Month 6**				
GMT	8.0 (6.6–9.8)	28.8 (22.3–37.2)	7.1(4.8–10.5)	<0.0001
GMT fold increase	1.6 (1.3–2.0) ^4^	5.8 (4.5–7.4) ^5^	1.4 (1.0–2.1) ^8^	<0.0001
**D614G**				
**Baseline**				
GMT	5 (5–5)	5 (5–5)	5 (5–5)	
**Day 21/Day 28**				
GMT	9.0 (7.5–11.0)	9.0 (7.3–11.2)	5.2 (4.8–5.5)	0.050
GMT fold increase	1.8 (1.5–2.2)	1.8 (1.5–2.2)	1.0 (1.0–1.1)	0.050
**D614G**				
**Day 56**				
GMT	13.1 (10.7–16.1)	81.4 (60.0–110.5)	11.8 (9.6–14.5)	<0.0001
GMT fold increase	2.6 (2.1–3.2)	16.3 (12.0–22.1)	2.4 (1.9–2.9)	<0.0001
**Month 6**				
GMT	5.7 (5.0–6.4)	21.5 (17.0–27.2)	5 (5–5)	<0.0001
GMT fold increase	1.1 (1.0–1.3)	4.3 (3.4–5.4)	1.0 (1.0–1.0)	<0.0001
**Alpha**				
**Baseline**				
GMT	5 (5–5)	5 (5–5)	5 (5–5)	
**Day 21/Day 28**				
GMT	7.5 (6.1–9.2)	9.0 (7.1–11.5)	5.4 (4.9–5.9)	0.177
GMT fold increase	1.5 (1.2–1.8)	1.8 (1.4–2.3)	1.1 (1.0–1.2)	0.177
**Day 56**				
GMT	11.5 (8.9–14.7)	146.7 (106.2–202.7)	11.6 (9.0–15.0)	<0.0001
GMT fold increase	2.3 (1.8–2.9)	29.3 (21.2–40.5)	2.3 (1.8–3.0)	<0.0001
**Month 6**				
GMT	6.6 (5.5–7.8)	37.9 (27.7–51.8)	7.1 (4.8–10.5)	<0.0001
GMT fold increase	1.3 (1.1–1.6)	7.6 (5.5–10.4)	1.4 (1.0–2.1)	<0.0001
**Beta**				
**Baseline**				
GMT	5 (5–5)	5 (5–5)	5 (5–5)	
**Day 21/Day 28**				
GMT	5 (5–5)	5 (5–5)	5.2 (4.8–5.5)	0.366
GMT fold increase	1.0 (1.0–1.0)	1.0 (1.0–1.0)	1.0 (1.0–1.1)	0.366
**Day 56**				
GMT	5.4 (4.8–5.9)	15.7 (12.0–20.5)	5 (5–5)	<0.0001
GMT fold increase	1.1 (1.0–1.2)	3.1 (2.4–4.1)	1.0 (1.0–1.0)	<0.0001
**Month 6**				
GMT	5 (5–5)	12.2 (9.8–15.2)	5 (5–5)	<0.0001
GMT fold increase	1.0 (1.0–1.0)	2.4 (2.0–3.0)	1.0 (1.0–1.0)	<0.0001
**Theta**				
**Baseline**				
GMT	5 (5–5)	5 (5–5)	5 (5–5)	
**Day 21/Day 28**				
GMT	5.4 (5.0–5.7)	6.7 (6.0–7.6)	5.2 (4.8–5.5)	0.001
GMT fold increase	1.1 (1.0–1.1)	1.3 (1.2–1.5)	1.0 (1.0–1.1)	0.001
**Day 56**				
GMT	5.9 (5.2–6.8)	39.3 (27.6–55.9)	7.1 (5.9–8.6)	<0.0001
GMT fold increase	1.2 (1.0–1.4)	7.9 (5.5–11.2)	1.4 (1.2–1.7)	<0.0001
**Theta**				
**Month 6**				
GMT	5.3 (4.9–5.6)	14.4 (11.5–18.1)	5 (5–5)	<0.0001
GMT fold increase	1.1 (1.0–1.1)	2.9 (2.3–3.6)	1.0 (1.0–1.0)	<0.0001
**Delta**				
**Baseline**				
GMT	5 (5–5)	5 (5–5)	5 (5–5)	
**Day 21/Day 28**				
GMT	5.9 (5.3–6.6)	8.6 (7.1–10.3)	5.7 (5.0–6.5)	0.008
GMT fold increase	1.2 (1.1–1.3)	1.7 (1.4–2.1)	1.1 (1.0–1.3)	0.008
**Day 56**				
GMT	7.4 (6.2–8.8)	74.6 (55.4–100.7)	8.0 (6.6–9.8)	<0.0001
GMT fold increase	1.5 (1.2–1.8)	14.9 (11.1–20.1)	1.6 (1.3–2.0)	<0.0001
**Month 6**				
GMT	5.5 (5.0–6.1)	22.7 (17.3–29.9)	5 (5–5)	<0.0001
GMT fold increase	1.1 (1.0–1.2)	4.5 (3.5–6.0)	1.0 (1.0–1.0)	<0.0001

^1^: data are mean values (95% CI). ^2^: the blood samples of the BNT162 group collected on day 21. ^3^: 41 individuals collected blood samples on day 28. ^4^: 28 individuals collected blood samples at 6 months. ^5^: 38 individuals collected blood samples at 6 months. ^6^: 40 individuals collected blood samples on day 28. ^7^: 38 individuals collected blood samples on day 56. ^8^: 4 individuals collected blood samples at 6 months. ^9^: *p* < 0.05, the results are significantly different.

**Table 3 vaccines-10-00072-t003:** Surrogate NAb and anti-N IgG.

	Combo (*n* = 42)	BNT162b2 (*n* = 40)	CoronaVac (*n* = 41)	*p* ^9^
**Surrogate NAb** **Mean (AU/mL) ^1^**				
Baseline	4.1 (3.9–4.3)	4.0 (4.0–4.0)	4.1 (3.9–4.4)	0.610
Day21 ^2^/Day28	37.1 (27.6–46.6) ^3^	70.6 (30.3–110.8)	5.5 (4.7–6.4) ^6^	0.001
Day 56	40.5 (24.6–56.4)	594.7 (509.9–679.5)	38.8 (27.9–49.6) ^7^	<0.0001
Month 6	8.6 (7.2–9.9) ^4^	138.5 (80.5–196.6) ^5^	9.2 (3.5–14.9) ^8^	0.001
**Anti-N IgG** **Mean (AU/mL) ^1^**				
Baseline	1.4 (1.0–1.9)	1.7 (0.6–2.8)	0.8 (0.6–1.0)	0.166
Day21 **^2^**/Day28	1.4 (0.9–1.8) ^3^	1.6 (0.6–2.6)	1.6 (1.1–2.0) ^6^	0.904
Day 56	1.4 (0.9–1.8)	1.3 (0.4–2.2)	21.8 (15.8–27.8) ^7^	<0.0001
Month 6	1.3 (0.7–1.9) ^4^	1.3 (0.4–2.2) ^5^	5.9 (-4.8–16.7) ^8^	0.007

^1^: data are mean values (95% CI). ^2^: the blood samples of the BNT162 group collected on day 21. ^3^: 41 individuals collected blood samples on day 28. ^4^: 28 individuals collected blood samples at 6 months. ^5^: 38 individuals collected blood samples at 6 months. ^6^: 40 individuals collected blood samples on day 28. ^7^: 38 individuals collected blood samples on day 56. ^8^: 4 individuals collected blood samples at 6 months. ^9^: *p* < 0.05, the results are significantly different.

**Table 4 vaccines-10-00072-t004:** Adverse events.

	Combo (*n* = 42)	BNT162b2 (*n* = 33)	CoronaVac (*n* = 35)	*p* ^2^
**Post first dose** **System reactions**	19 (45.2%)	15 (45.5%)	13 (37.1%)	0.761
Fever	2 (4.8%)	1 (3.0%)	0 (0)	0.636
Chills	0 (0)	0 (0)	2 (5.7%)	0.187
Headache	6 (14.3%)	6 (18.2%)	4 (11.4%)	0.702
Tiredness	12 (31.0%)	11 (33.3%)	9 (25.7%)	0.807
Nausea	1 (2.4%)	1 (3.0%)	3 (8.6%)	0.516
Vomit	0 (0)	0 (0)	0 (0)	-
Diarrhea	1 (2.4%)	2 (6.1%)	2 (5.7%)	0.732
Muscle pain	7 (16.7%)	9 (27.3%)	6 (17.1%)	0.500
Joint pain	1 (2.4%)	4 (12.1%)	2 (5.7%0	0.255
Skin rash	3 (7.1%)	1 (3.0%)	1 (2.9%)	0.624
SAE ^1^	0 (0)	0 (0)	0 (0)	-
**Local symptoms**	41 (97.6%)	27 (81.8%)	12 (34.3%)	<0.0001
Pain	41 (97.6%)	25 (75.8%)	12 (34.3%)	<0.0001
Redness	3 (7.1%)	7 (21.2%)	0 (0)	0.005
Swelling	6 (14.3%)	12 (36.4%)	0 (0)	<0.0001
Ecchymosis	4 (9.5%)	3 (9.1%)	0 (0)	0.200
Itching	2 (4.8%)	3 (9.1%)	1 (2.9%)	0.200
**Post second dose** **System reactions**	13 (31.0%)	22 (66.7%)	8 (22.9%)	<0.0001
Fever	0 (0)	2 (6.1%)	0 (0)	0.088
Chills	0 (0)	6 (18.2%)	1 (2.9%)	0.002
Headache	6 (14.3%)	9 (27.3%)	3 (8.6%)	0.107
Tiredness	10 (23.8%)	17 (51.5%)	3 (8.6%)	<0.0001
Nausea	2 (4.8%)	5 (15.2%)	2 (5.7%)	0.233
Vomit	0 (0)	1 (3.0%)	1 (2.9%)	0.524
Diarrhea	0 (0)	3 (9.1%)	3 (8.6%)	0.157
Muscle pain	3 (7.1%)	16 (48.5%)	3 (8.6%)	<0.0001
Joint pain	1 (2.4%)	5 (15.2%)	1 (2.9%)	0.046
Skin rash	0 (0)	1 (3.0%)	0 (0)	0.300
SAE **^1^**	0 (0)	0 (0)	0 (0)	-
**Local symptoms**	15 (35.7%)	25 (75.8%)	7 (20.0%)	<0.0001
Pain	15 (35.7%)	23 (69.7%)	7 (20.0%)	<0.0001
Redness	2 (4.8%)	9 (27.3%)	1 (2.9%)	0.001
Swelling	2 (4.8%)	11 (33.3%)	0 (0)	<0.0001
Ecchymosis	0 (0)	1 (3.0%)	0 (0)	0.300
Itching	0 (0)	5 (15.2%)	0 (0)	0.002

^1^: SAE: severe adverse events, vaccine-related undesired events including death, disability or life-threatening conditions. ^2^: *p* < 0.05, the results are significantly different.

## Data Availability

The data used to support the findings of this study are included within the article.

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
