# Peer review of "Immunogenicity of a Heterologous Prime-Boost COVID-19 Vaccination with mRNA and Inactivated Virus Vaccines Compared with Homologous Vaccination Strategy against SARS-CoV-2 Variants"

_vaccines, 2022, doi:10.3390/vaccines10010072_

Round 1

Reviewer 1 Report

Zhang et al report the immunogenicity of a heterologous prime-boost COVID-19 vaccination with BNT162b2 (an mRNA-based vaccine) and CoronaVac (an inactivated virus based vaccine) and compared with those with homologous vaccination. The results showed that both heterologous and homologous vaccination strategies were safe, the immunogenicity using homologous BNT162b2 was higher than both homologous CoronaVac and heterologous with BNT162b2 followed by CoronaVac. The neutralization against SARS-CoV-2 waned at 6 months in all three groups. Overall the manuscript is well-written. However, authors should do some more analysis on their data. The results clearly showed that the significant difference of immunogenicity among the three arms of the vaccination strategies, but the authors should also carry out a post hoc analysis to see if there are any differences between groups (especially between the homologous CoronaVac group and the group of BNT162b2 followed by CoronaVac. This analysis will help to draw the conclusion if the heterologous combination between BNT162b2 and CoronaVac will improve or have no significant impact on the immunogenicity against SARS-CoV-2. By looking at the data themselves, it seems for most of the cases, there is no significant impact between the homologous CoronaVac group and the combo group, but the post hoc analysis between groups will provide a clear picture. Based on this analysis, the authors should also discuss the utility of the heterologous strategy with BNT162b2 followed by the CoronaVac. The immune response is not known if the heterologous group with the CoronaVac first, followed by BNT162b2, and therefore it is not known about the impact of BNT162b2 as a booster on those people vaccinated with CoronaVac. Authors should include that as part of the limitation of the study.

Minor issues:

line 32, what does “DV” mean?

Table 2, superscript 9 should be 8? Since that corresponds to footnote 8, and footnote 9 is regarding p-value.

Table 3: the superscripts in the table are alphabetical, while the footnotes are numeric.

Figure 2: explain the p-value between which groups.

Author Response

RE: A point-to-point response to Reviewer’s comments

“Immunogenicity of a heterologous prime-boost COVID-19 vaccination with mRNA and inactivated virus vaccines compared with homologous vaccination strategy against SARS-CoV-2 variants” (vaccines-1500201) by Ruiqi Zhang, Danlei Liu, Ka-Yi Leung, Yujing Fan, Lu Lu, Pui-Chun Chan, Kelvin Kai-Wang To, Honglin Chen, Kwok-Yung Yuen, Kwok-Hung Chan and Ivan Fan-Ngai Hung

We are grateful to the helpful comments given by referee. We have carefully revised the manuscript accordingly. A point-to-point response to reviewers’ comments is given as follow.

  1. Zhang et al report the immunogenicity of a heterologous prime-boost COVID-19 vaccination with BNT162b2 (an mRNA-based vaccine) and CoronaVac (an inactivated virus based vaccine) and compared with those with homologous vaccination. The results showed that both heterologous and homologous vaccination strategies were safe, the immunogenicity using homologous BNT162b2 was higher than both homologous CoronaVac and heterologous with BNT162b2 followed by CoronaVac. The neutralization against SARS-CoV-2 waned at 6 months in all three groups. Overall the manuscript is well-written. However, authors should do some more analysis on their data. The results clearly showed that the significant difference of immunogenicity among the three arms of the vaccination strategies, but the authors should also carry out a post hoc analysis to see if there are any differences between groups (especially between the homologous CoronaVac group and the group of BNT162b2 followed by CoronaVac. This analysis will help to draw the conclusion if the heterologous combination between BNT162b2 and CoronaVac will improve or have no significant impact on the immunogenicity against SARS-CoV-2. By looking a t the data themselves, it seems for most of the cases, there is no significant impact between the homologous CoronaVac group and the combo group, but the post hoc analysis between groups will provide a clear picture. Based on this analysis, the authors should also discuss the utility of the heterologous strategy with BNT162b2 followed by the CoronaVac. The immune response is not known if the heterologous group with the CoronaVac first, followed by BNT162b2, and therefore it is not known about the impact of BNT162b2 as a booster on those people vaccinated with CoronaVac. Authors should include that as part of the limitation of the study.

Response: We thank Reviewer for the comments. We perform the post hoc test to compare the immunogenicity of two groups, especially CoronaVac group and heterologous BNT162b2-CoronaVac group. We found that there was no statistically significant difference in GMT at day 56 and month 6 between heterologous BNT162b2-CoronaVac group and CoronaVac group, and the results were presented on Figure 2. We have added the results of post hoc multiple comparison to Results section.

Line 173-176: On day 56, GMT in BNT162b2 group was significantly higher than that in Combo (p<0.0001) and CoronaVac groups (p<0.0001), while no statistically significant difference was observed in GMT between Combo and CoronaVac groups (Figure 2a).

Line 185-191: After full vaccination, there was no statistically significant difference in GMT at day 56 and month 6 between Combo and CoronaVac groups (Figure 2b). For alpha variant, in-dividuals in BNT162b2 group showed higher GMT on day 56 (146.7) (p<0.0001) and month 6 (37.9) (p<0.0001) than these from Combo group (day 56 GMT, 11.5; month 6 GMT, 6.6) and CoronaVac group (day 56 GMT, 11.6; month 6 GMT, 7.1), while Combo group showed comparable antibody response at day 56 and month 6 with CoronaVac group (Table 2) (Figure 2c).

Line 194-208: The GMT on day 56 in BNT162b2 group was higher than that in Combo (p<0.0001) and CoronaVac (p<0.0001) groups, and no statistically significant difference was observed in GMT between Combo and CoronaVac groups (Figure 2d). These vaccination strategies also showed low effectiveness against theta variant, and the day 56 GMT in Combo group was 5.9 (95% CI, 5.2-6.8) which was lower than that in BNT162b2 (day 56 GMT, 39.3; 95% CI, 27.6-55.9) (p<0.0001) and comparable with CoronaVac group (day 56 GMT, 7.1; 95%CI, 5.9-8.6) (Table 2) (Figure 2e). For delta variant, the day 56 GMT and month 6 GMT in BNT162b2 platform (day 56, 74.6; month 6, 13) was higher than that in Combo (day 56, 7.4; month 6, 5.5) and CoronaVac platforms (day 56, 8.0; month 6, 5) (p<0.0001) (Table 2)(Figure 2f). When compared the immunogenicity against delta between Combo and CoronaVac group, no statistically significant difference was observed at day 56 and month 6 (Figure 2f). Overall, BNT162b2 platform showed highest immunogenicity, and Combo platform induced comparable antibody response with CoronaVac groups against WT and five variants.

Based on the comparison of homologous CoronaVac group and heterologous BNT162b2-CoronaVac group, we have added the conclusion that the antibody response against WT and variants induced by Combo vaccination strategy was comparable to that in CoronaVac group after full vaccination in Conclusion section.

Line 394-397: The overall vaccine immunogenicity of the Combo platform at day 56 and 6 months was inferior to the homologous BNT162b2 platform but comparable to the homologous CoronaVac platform, against both the WT and all variants.

We have added the utility of heterologous BNT162b2-CoronaVac platform in Discussion section.

Line 309-318: Studies on the mRNA vaccines have demonstrated its high clinical effectiveness and are adopted widely in the world [24, 25]. However, the shortage of mRNA COVID-19 vac-cines in some countries resulting in the delay in the second dose [26]. For these indi-viduals who cannot receive the booster dose of BNT162b2 on time, they can opt for one dose of CoronaVac to complete the vaccination regime. Furthermore, subjects who cannot tolerate the side effects induced by booster dose of BNT162b2, can also switch to one dose of CoronaVac after priming with the BNT162b2 (Table 4).

The results of antibody response in individuals receiving a dose of CoronaVac as primer and one dose of BNT162b2 as booster was lacking in the study, and we have added it as limitation in Discussion section.

Line 389-391: Besides, the immunogenicity of CoronaVac as primer followed by a booster of BNT162b2 vaccine is deserved to investigate, and we have started to recruit volunteers to receive heterologous CoronaVac-BNT162b2 vaccination strategy.  

Minor issues:

  1. line 32, what does “DV” mean?

Response: We thank Reviewer for the comments. We have changed the DV to delta variant.

Line 31: “delta variant”

  1. Table 2, superscript 9 should be 8? Since that corresponds to footnote 8, and footnote 9 is regarding p-value.

Response: We thank Reviewer for the comments. We have corrected it and changed the superscript 9 to 8 in Table 2.

Table 3: the superscripts in the table are alphabetical, while the footnotes are numeric.

Response: We thank Reviewer for the comments, and we have deleted the superscript letter.

Figure 2: explain the p-value between which groups.

Response: We thank Reviewer for the comments. In figure 2, we performed post hoc multiple comparison to analyse the difference in immunogenicity between two groups, and we showed p value on the graph if the value was less than 0.05 which represents statistically significant difference.

Line 232-233: To analyse the difference in immunogenicity between two groups, post hoc multiple comparison was performed and p value was showed on the graph if the value was less than 0.05 which represents statistically significant difference.

Reviewer 2 Report

In this manuscript, Hung and colleagues compared the SARS-CoV2 vaccine efficacies of three protocols, mRNA vaccine, inactivation vaccine, and their combination. They determined neutralizing antibody titers not only to WT virus but also to variants. They also evaluate the safety of these protocols. Because COVID19 world-wide pandemic is still on going and new variants are appearing, the information described in this manuscript is worth publishing.
However, by their study design, it is difficult to correctly assess the boost immunization with CoronaVac after primary immunization by BNT162b2. Actually, I could not find any benefit of boost immunization in the Combo group. In Table 2, GMT of the Combo group was 23.7 at day 28 and it did not increase significantly after boosting at day 56. If GMT at day 56 after BMT162b2 vaccination without boosting was less than day 56 GMT of Combo group, boost immunization with CoronaVac would be beneficial. If booster immunization by Corona Vac does not increase NAb titer, it should not be done even if no sever adverse event was observed. The author should provide additional data or information to prove boost immunization is really effective to prevent COVID-19.
In regards to adverse events in Table 4, it is not clear that adverse events were counted only after booster immunization, or all events were counted. In order to assess the adverse events of heterologous vaccination correctly, it is better to count the events before and after boosting separately.
I suggest that the authors re-write the paper considering the above.

Minor mistakes:
line 126, CO2 -> 2 in subscript
line 131, geometric mean tire -> titer
line 220, Nab -> NAb
line 224, anti- N -> anti-N
Table 2 and Table 3, Numbering of annotation by 1, 2, 3,  or a, b, c,  ?

Author Response

RE: A point-to-point response to Reviewer’s comments

“Immunogenicity of a heterologous prime-boost COVID-19 vaccination with mRNA and inactivated virus vaccines compared with homologous vaccination strategy against SARS-CoV-2 variants” (vaccines-1500201) by Ruiqi Zhang, Danlei Liu, Ka-Yi Leung, Yujing Fan, Lu Lu, Pui-Chun Chan, Kelvin Kai-Wang To, Honglin Chen, Kwok-Yung Yuen, Kwok-Hung Chan and Ivan Fan-Ngai Hung

We are grateful to the helpful comments given by referee. We have carefully revised the manuscript accordingly. A point-to-point response to reviewers’ comments is given as follow.

  1. In this manuscript, Hung and colleagues compared the SARS-CoV2 vaccine efficacies of three protocols, mRNA vaccine, inactivation vaccine, and their combination. They determined neutralizing antibody titers not only to WT virus but also to variants. They also evaluate the safety of these protocols. Because COVID19 world-wide pandemic is still on going and new variants are appearing, the information described in this manuscript is worth publishing.
    However, by their study design, it is difficult to correctly assess the boost immunization with CoronaVac after primary immunization by BNT162b2. Actually, I could not find any benefit of boost immunization in the Combo group. In Table 2, GMT of the Combo group was 23.7 at day 28 and it did not increase significantly after boosting at day 56. If GMT at day 56 after BMT162b2 vaccination without boosting was less than day 56 GMT of Combo group, boost immunization with CoronaVac would be beneficial. If booster immunization by Corona Vac does not increase NAb titer, it should not be done even if no sever adverse event was observed. The author should provide additional data or information to prove boost immunization is really effective to prevent COVID-19. In regards to adverse events in Table 4, it is not clear that adverse events were counted only after booster immunization, or all events were counted. In order to assess the adverse events of heterologous vaccination correctly, it is better to count the events before and after boosting separately.
    I suggest that the authors re-write the paper considering the above.

Response: We thank Reviewer for the comments. We agree with the reviewer that it is difficult to assess whether the boost immunization with second dose CoronaVac is effective in preventing COVID-19 infection, as there were very few imported COVID-19 patients and no local patient in Hong Kong over the previous 6 months, due to the very strict quarantine policy imposed on the returned travelers. We have added this to the limitation.

Despite an inferior neutralizing antibody response when compared to the homologous BNT162b2, the booster dose of CoronaVac vaccine could maintain the immune response induced by the primer dose of BNT162b2 in the Combo group. Studies on the mRNA vaccines have demonstrated its high clinical effectiveness and are adopted widely in the world [1,2]. However, the shortage of mRNA COVID-19 vaccines in some countries resulting in the delay in the second dose [3]. For these individuals who cannot receive the booster dose of BNT162b2 on time, they can opt for one dose of CoronaVac to complete the vaccination regime. Furthermore, subjects who cannot tolerate the side effects induced by booster dose of BNT162b2, can also switch to one dose of CoronaVac after priming with the BNT162b2.

We have added this to the Discussion.

Line 298-309: In one clinical study, Vasileiou et al. reported that the effectiveness of primer-dose BNT162b2 was 85% at day 28-34 post-vaccination, and then dropped to a level of 64% after 42 days post-vaccination [22]. Furthermore, from our previous study, the neutral-izing activity in recovered patients with previous COVID-19 infection decreased sig-nificantly at week 6 after discharge [23]. Thus, the antibody response on day 56 might be lower than on day 28 in healthy individuals who have just received one dose of BNT162b2. In this study, the day 56 GMT was comparable to the day 28 GMT in the Combo group. Despite an inferior neutralizing antibody response when compared to the homologous BNT162b2, the, booster dose of CoronaVac vaccine could maintain the immune response induced by the primer dose of BNT162b2 in the Combo group.

We have amended Table 4 accordingly, and counted the frequency of adverse events after first dose and second dose separately. We also revised the Results section.

Line 259-273: After primer dose of vaccine, the incidence of total local symptoms in BNT162b2 and Combo groups were significantly higher than that of CoronaVac (Combo, 97.6%; BNT162b2, 81.8%; CoronaVac, 34.3%; p< 0.0001), and there was no significant difference in total systemic reactions between three groups (Combo, 45.2%; BNT162b2, 45.5%; CoronaVac, 37.1%; p=0.761). The dose 2 vaccine induced higher frequencies of systemic reactions (66.7%) (p<0.0001) and local reactions (75.8%) (p<0.0001) in BNT162b2 group than these in Combo (systemic reactions, 31.0%; local reactions, 35.7%) and CoronaVac groups (systemic reactions, 22.9%; local reactions, 20.0%) (Table 4). The most common symptom induced by a booster in three groups was injection site pain (Combo, 35.7%; BNT162b2, 69.7%; CoronaVac, 20.0%). After the second dose, the frequencies of chill, tiredness, muscle pain, joint pain, redness, swelling, and itching in BNT162b2 group were higher than these in Combo and CoronaVac groups (Table 4). No SAE was observed in the study after vaccination.

We have revised Results and Discussion section according to Reviewer’s advice.

Minor mistakes:
2. line 126, CO2 -> 2 in subscript

Response: We thank Reviewer for the comments, and we have corrected it.
Line 115: CO2

line 131, geometric mean tire -> titer

Response: We thank Reviewer for the advice and we have changes it.

Line 130: titre

line 220, Nab -> NAb

Response: We thank Reviewer for the comments, and we have corrected it.

Line 237: NAb

line 224, anti- N -> anti-N

Response: We thank Reviewer for the comments, and we have deleted the space.

Line 244: anti-N

Table 2 and Table 3, Numbering of annotation by 1, 2, 3,  or a, b, c, ?

Response: We thank Reviewer for the comments, and the superscribes in tables are numeric and we have corrected that accordingly.

Reference:

  1. Kon, E.; Elia, U.; Peer, D. Principles for designing an optimal mRNA lipid nanoparticle vaccine. Curr Opin Biotechnol 2021, 73, 329-336, doi:10.1016/j.copbio.2021.09.016.
  2. Thomas, S.; Moreira Jr., E.D.; Kitchin, N.; Absalon, J.; Gurtman, A.; Lockhart, S.; Perez, J.L.; Marc, G.P.; Fernando P. Polack, F.P.; Zerbini, C.; et al. Six Month Safety and Efficacy of the BNT162b2 mRNA COVID-19 Vaccine. preprint 2021, 21, doi.org/10.1101/2021.07.28.21261159.
  3. Miron, O.; Wilf-Miron, R.; Davidovitch, N. Effectiveness of COVID-19 vaccines BNT162b2 and mRNA-1273 by days from vaccination: A reanalysis of clinical trial data. preprint 2021, 14, doi:org/10.2139/ssrn.3791560.

Round 2

Reviewer 1 Report

Given the booster definition for COVID-19 vaccines, the authors should change the “booster shot” in the ms to “second dose of the COVID-19 vaccine, and reserve the “booster” as the booster after the two-dose regimen of COVID-19 vaccines. Further, it is also worthy to mention the importance of investigating the mix-and-match approach for the booster shot (following the two-does regimen).

Author Response

RE: A point-to-point response to Reviewer’s comments

“Immunogenicity of a heterologous prime-boost COVID-19 vaccination with mRNA and inactivated virus vaccines compared with homologous vaccination strategy against SARS-CoV-2 variants” (vaccines-1500201) by Ruiqi Zhang, Danlei Liu, Ka-Yi Leung, Yujing Fan, Lu Lu, Pui-Chun Chan, Kelvin Kai-Wang To, Honglin Chen, Kwok-Yung Yuen, Kwok-Hung Chan and Ivan Fan-Ngai Hung

We are grateful to the helpful comments given by referee. We have carefully revised the manuscript accordingly. A point-to-point response to reviewers’ comments is given as follow.

  1. Given the booster definition for COVID-19 vaccines, the authors should change the “booster shot” in the ms to “second dose of the COVID-19 vaccine, and reserve the “booster” as the booster after the two-dose regimen of COVID-19 vaccines. Further, it is also worthy to mention the importance of investigating the mix-and-match approach for the booster shot (following the two-does regimen)

Response: We thank Reviewer for the advice. We have changed the “booster dose” accordingly and added the importantce of investigation of the mix-and-match approach for the booster shot in people after full vaccination.

Line 101-102: second doses of COVID-19 vaccines

Line 266: the second dose of COVID-19 vaccine

Line 303: one dose of CoronaVac vaccine

Line 307-308: second dose of BNT162b2 vaccine

Line 310: second dose of the BNT162b2 vaccine

Line 340: second dose of the BNT162b2 vaccine

Line 354-355: booster dose should be administered to these individuals receiving BNT162b2 at 5 months post-second dose

Line 359-360: third dose of vaccine

Reviewer 2 Report

According to the referees' comments, the authors revised the manuscript. Now, the differences of three vaccine strategies were described more clearly and the readers can estimate efficacies more correctly. The limitation of the study was also adequately described.

I do not find further problems and I think that the revised manuscript is suitable for publication. 

Author Response

RE: A point-to-point response to Reviewer’s comments

“Immunogenicity of a heterologous prime-boost COVID-19 vaccination with mRNA and inactivated virus vaccines compared with homologous vaccination strategy against SARS-CoV-2 variants” (vaccines-1500201) by Ruiqi Zhang, Danlei Liu, Ka-Yi Leung, Yujing Fan, Lu Lu, Pui-Chun Chan, Kelvin Kai-Wang To, Honglin Chen, Kwok-Yung Yuen, Kwok-Hung Chan and Ivan Fan-Ngai Hung

We are grateful to the helpful comments given by referee. We have carefully revised the manuscript accordingly. A point-to-point response to reviewers’ comments is given as follow.

1. According to the referees' comments, the authors revised the manuscript. Now, the differences of three vaccine strategies were described more clearly and the readers can estimate efficacies more correctly. The limitation of the study was also adequately described.

I do not find further problems and I think that the revised manuscript is suitable for publication. 

Response: We thank Reviewer for the comments.
